Altered vaginal cervical microbiota diversity contributes to HPV-induced cervical cancer via inflammation regulation

Yang Yiheng 1
Zhu Jufan 1
Feng Renqian 1
Han Mengfei 1
Chen Fenghua 2
Hu Yan 1 drhuyan@wzhospital.cn
1 Department of Obstetrics and Gynecology, The First Affiliated Hospital of Wenzhou Medical University , Wenzhou , China
2 Yiwu Central Hospital , Yiwu , China
Tandon Ravi
Electronic publication date: 2024 Jun 12
Publication date: 2024
Volume: 12
Electronic Location ID: e17415
Received 2023 Nov 3; Accepted 2024 Apr 28
Copyright: © 2024 Yang et al.
Copyright year: 2024
Copyright holder: Yang et al.
License: This is an open access article distributed under the terms of the Creative Commons Attribution License, which permits unrestricted use, distribution, reproduction and adaptation in any medium and for any purpose provided that it is properly attributed. For attribution, the original author(s), title, publication source (PeerJ) and either DOI or URL of the article must be cited.
License URL: https://creativecommons.org/licenses/by/4.0/

Keywords: 16S rRNA gene, HPV, Squamous intraepithelial lesions, Vaginal microbiota

Funding: Department of Zhejiang Provincial Natural Science Foundation of China Y19H040024 This work was supported by the Department of Zhejiang Provincial Natural Science Foundation of China (Grant Number Y19H040024). The funders had no role in study design, data collection and analysis, decision to publish, or preparation of the manuscript.

==============================
Background

Cancer has surpassed infectious diseases and heart ailments, taking the top spot in the disease hierarchy. Cervical cancer is a significant concern for women due to high incidence and mortality rates, linked to the human papillomavirus (HPV). HPV infection leads to precancerous lesions progressing to cervical cancer. The cervix’s external os, near the vagina, hosts various microorganisms. Evidence points to the link between vaginal microbiota and HPV-induced cervical cancer. Cervical cancer onset aligns with an imbalanced Th1/Th2 immune response, but the role of vaginal microbiota in modulating this imbalance is unclear.

Methods

In this study, we collected vaginal samples from 99 HPV-infected patients across varying degrees of lesions, alongside control groups. These samples underwent bacterial DNA sequencing. Additionally, we employed Elisa kits to quantify the protein expression levels of Th1/Th2 cytokines IL2, IL12, IL5, IL13, and TNFa within the centrifuged supernatant of vaginal-cervical secretions from diverse research subjects. Subsequently, correlation analyses were conducted between inflammatory factors and vaginal microbiota.

Results

Our findings highlighted a correlation between decreased Lactobacillus and increased Gardenerella presence with HPV-induced cervical cancer. Functionally, our predictive analysis revealed the predominant enrichment of the ABC transporter within the vaginal microbiota of cervical cancer patients. Notably, these microbiota alterations exhibited correlations with the production of Th1/Th2 cytokines, which are intimately tied to tumor immunity.

Conclusions

This study suggests the potential involvement of vaginal microbiota in the progression of HPV-induced cervical cancer through Th1/Th2 cytokine regulation. This novel insight offers a fresh perspective for early cervical cancer diagnosis and future prevention strategies.

Introduction

Cervical cancer ranks among the most prevalent malignancies affecting women worldwide, and its primary instigator is persistent human papillomavirus (HPV) infection. Within the papillomavirus family, HPV constitutes a subset of double-stranded DNA viruses (Jain et al., 2023). Though the majority of HPV-infected women can eliminate the virus via autoimmunity, an enduring 5% to 10% are unable to do so. This prolonged HPV presence can culminate in cervical precancerous lesions and eventual cervical cancer (Tsang et al., 2020). Hence, investigating the pathogenesis of HPV-induced cervical cancer emerges as an imperative pursuit.

Healthy women’s vaginal microbiota, host, and environment maintain a dynamic balance, intricately regulating each element in a symbiotic coexistence (Rodriguez-Cerdeira, Sanchez-Blanco & Alba, 2012). In healthy women, the vaginal microbiota encompasses over 50 microorganisms, with Lactobacillus dominating as the most abundant and significant member. Pioneering the classification of vaginal microbiota through high-throughput sequencing technology, (Ravel et al., 2011; Nunn & Forney, 2016). delineated five common microbiota types, four of which are primarily Lactobacillus-dominant. This Lactobacillus contingent safeguards vaginal health by producing lactic acid, hydrogen peroxide (H2O2), bacteriocins, and other compounds. Conversely, the fifth type, marked by an abundance of anaerobic bacteria and less Lactobacillus, typically raises vaginal pH above 4.5, indicative of bacterial vaginosis (Ravel et al., 2011). Among these, Lactobacillus crispatus and Lactobacillus inerts are most prevalent, trailed by Lactobacillus gasseri and Lactobacillus johnsonii (Zhang et al., 2020). The metabolites of diverse microorganisms, including hydrogen peroxide, bacteriocins, and defensins (a type of protein with antimicrobial activity that can disrupt the cell membrane of microorganisms), exhibit inhibitory effects on other bacteria, thus maintaining a delicate balance that fortifies the female vaginal-cervical milieu against external microorganisms and upholding a biological barrier function (Gong et al., 2014).

Studies have reported a close association between the female cervical vaginal microbiota and human papillomavirus (HPV) infection, cervical intraepithelial neoplasia (CIN), and the development of cervical cancer (Chen et al., 2020; Gupta, Kakkar & Bhushan, 2019; Mitra et al., 2020). Vaginal microbiota diversity is linked to HPV infection rates, persistence, and the severity of CIN (Mahajan et al., 2022; Mitra et al., 2020; Stapleton et al., 2021; Yu et al., 2020). Specific bacterial species within the microbiota could influence the local immune response, potentially impacting the progression of HPV-related conditions (Lin et al., 2020; Sims, Colbert & Klopp, 2021). Notably, Sneathia has been significantly enriched in CIN samples, and its presence is associated with altered expression of immune factors (Łaniewski et al., 2018). Fusobacterium has been found to correlate with elevated expression of IL-4 and transforming growth factor (TGF)-β1, suggesting its role in immunosuppression within the microenvironment of invasive cervical cancer (ICC) (Audirac-Chalifour et al., 2016). Furthermore, past studies indicate an association between cervical cancer and Th1/Th2 imbalance (Clerici, Shearer & Clerici, 1998; Lin et al., 2019), however, it remains unclear whether changes in vaginal microbiota play a role in regulating this imbalance. This study collected vaginal samples from 99 women for bacterial DNA sequencing, including a non-HPV infected control group (N group), an HPV persistent infection group (HPV group), a cervical low-grade squamous epithelial lesion group (LSIL group), a cervical high-grade squamous epithelial lesion group (HSIL group), and a cervical squamous cell carcinoma group (CA group). The aim was to investigate changes in the vaginal microbiota associated with HPV-induced cervical cancer and precancerous lesions, and to analyze the correlation between vaginal microbiota and Th1/Th2 factors. This research provides a new perspective on the potential involvement of vaginal microbiota in HPV-induced cervical cancer.

Methods

Written informed consent was obtained from the individual for the publication of any potentially identifiable images or data included in this article.

This study was conducted in accordance with the Declaration of Helsinki. The studies involving human participants were reviewed and approved by the Professional ethics committee of clinical research of the First Affiliated Hospital of Wenzhou Medical University (Approval No. YS2018-058). The participants provided their written informed consent to participate in this study.

Study population—inclusion and exclusion criteria

The research content of this study has undergone thorough scrutiny and received approval from our hospital’s Ethics Committee. The study population consisted of married women of childbearing age who sought medical care at our hospital’s gynecological cervix outpatient clinic between September 1, 2019, and February 29, 2020. Prior to participation, all research subjects were comprehensively informed about the research plan and the significance of this project, and their informed consent was duly obtained.

The enrolled participants were categorized into five groups based on their HR-HPV infection status and cervical pathology results. These groups include the HR-HPV persistent infection group (HPV group) without any abnormal cervical pathology findings, the cervical low-grade squamous epithelial lesion group (LSIL group), the cervical high-grade squamous epithelial lesion group (HSIL group), the cervical squamous cell carcinoma group (CA group), and a non-infected control group (H group).

All enrolled subjects needed to meet the following criteria simultaneously:

Participants were exclusively of local Han nationality and had no history of prolonged foreign residence.

No prior instances of cervical intraepithelial neoplasia, HPV infection, or cervical treatment were reported.

Participants were non-diabetic and free from immune system disorders, pregnancy, lactation, recent antibiotic or glucocorticoid use, and genitourinary or systemic infections.

Subjects were collected before any form of treatment, including drugs or surgery, was administered to the subjects in the five groups. Additionally, sexual activity, vaginal drug application, or any form of vaginal operation had not occurred within the preceding 3 days.

Specimen collection and processing

Collecting vaginal and cervical samples: The patient lies in a gynecological examination position. Using a sterile speculum to open the vagina, two disposable cotton swabs are gently rotated at least 10 times in the posterior fornix and around the cervix. After thoroughly absorbing vaginal secretions and cervical mucus, the swabs are carefully removed to avoid contamination from external genitalia and the vaginal opening. The disposable cotton swabs are then placed in a sterile, dry EP tube for extracting total bacterial DNA.

Sequencing of vaginal-cervical microbiota

Extraction of bacterial total DNA

In this experiment, the bacterial genome DNA extraction kit (spin column type) of BGI was used to extract total DNA from vaginal-cervical secretion swabs. The experimental steps were referred to its instructions.

After the total DNA of the samples was extracted, the purity and concentration of the extracted total DNA were determined by ultraviolet spectrophotometer (570 nm wavelength). The resulting product was electrophoresed on a 1% agarose gel at 150 V for 40 min to check the integrity of the sample and exclude RNA, protein and secondary metabolite contamination. DNA samples that have passed the electrophoresis test are stored in a −80 °C refrigerator.

16S rRNA-V4 region amplification (Reitmeier et al., 2020)

Using the total DNA of the sample as the template, the primers 514 F and 805 R were synthesized by Sanko. By consulting relevant literature, the 514 F primer sequence used in this experiment is GTGCCAGCMGCCGCGGTAA, and the 805 R primer sequence is GGACTACHVGGGTWTCTAAT, which can amplify the 16S rRNA-V4 region gene of the DNA sample. Use the kit provided by Taraka Company, and configure a 25 ul PCR reaction system according to the instructions: 10 × buffer (2.5 u L), dNTP (2 uL), Mg2 + (1.5 uL), P1 (0.5 uL), P2 (0.5 uL), Ex-taq (0.25 uL), template DNA (1.0 uL), H20 (16.75 uL). The PCR amplification conditions were selected as follows: pre-denaturation at 94 °C for 2 min, then denaturation at 94 °C for 30 s, annealing at 57 °C for 30 s, and maintenance at 72 °C for 30 s, a total of 30 cycles, and a final maintenance at 72 °C for 5 min. The PCR products were subjected to 1.0% agarose gel electrophoresis (voltage 180 V, electrophoresis time 20 min) to detect the yield and characteristics.

DNA sequencing

After amplification, the purified and qualified total PCR products were sent to Shanghai Bohao Biotechnology Co., Ltd. and sequenced by the GSFLX Titanuim (Roche, Basel, Switzerland) system.

Bioinformatics analysis

Sequencing data processing: After the original sequences obtained by high-throughput sequencing were split according to the sample Barcode tags, FLASH (version 1.2.3) software was used to splicing pairs of short reads (reads), and Cutadapt (version 1.9.1) software removes Barcode and primer sequences in the spliced sequence.

Usearch (version 8.1.1861) software performs preliminary quality control on sequences, and removes low-quality sequences, including sequences with ambiguous bases, primer errors, average quality score less than 25 or length less than 200 nt, The chimera sequences were checked and removed by using the de novo method of the usearch software, and finally obtained for clustering and classification analysis. The sequences were denoised by ussearch, and ASV representative sequences were obtained. The obtained representative sequences were compared and annotated with the RDP database using the method of QIIME package RDP Classifier, and the taxonomic information of each ASV corresponding species was obtained. Bacterial diversity analysis was based on the obtained ASV feature table, using pheatmap and ggplot2 in R (version 3.6.3) for visualization, and using the VennDiagram package to draw Venn diagrams to reveal ASVs shared by different groups.

Alpha and beta diversity: Use the Vegan package to calculate the diversity index of each sample, such as alpha diversity, Chao1 index, Shannon index, etc. The Vegan package calculates principal component analysis (PCoA) based on the bray_curtis distance matrix, and evaluates the β-diversity of samples and the similarity and difference of community composition among different groups.

The dplyr package was used to analyze the differences between groups, and Graphlan (version 0.9) software was used for visualization. Alpha diversity mainly analyzes species diversity in a single sample, which can reflect the species richness and species evenness of the microbial microbiota within a single sample (Stapleton et al., 2021).

LEfSe analysis (linear discriminant analysis coupled with effect size measurements).

The nonparametric Kruskal-Wallis rank sum test was first used to find species with significant differences in abundance between different groups, and then the Wilcoxon rank sum test was used to evaluate the accuracy of the species obtained in the previous step. LEfSe analysis is a new statistical method (Yu et al., 2020). LEfSe analysis can identify taxa, pathways, and genomic properties, discover and interpret high microbial markers, and help identify differences between two or two colonies.

Elisa test

In order to detect the vaginal-cervical immune microenvironment in research subjects in each group, we used Elisa kit to detect the protein expression level of Th1-related cytokines IL2, IL12, and Th2-related cytokines, IL5, IL13,and monocyte-macrophage-related factor TNFa in the centrifuged supernatant of vaginal-cervical secretions of different research subjects.

Statistical analysis

In this study, all data processing was done using SPSS 24.0 software. The multi-group data is tested by ANOVA analysis of variance, the comparison of two groups of data is by t test, and the analysis between each group is corrected by Bonferroni. Comparison of count data between groups was performed using χ2 test. The difference was considered statistically significant at P < 0.05.

Results

Patient cohort and characteristics

In this study, a total of 99 patients were meticulously selected and categorized into five distinct groups. Among these groups, four were composed of patients with HPV infection, and their classification was determined by the extent of cervical pathological changes. These four groups comprised the HR-HPV persistent infection group (HPV group) without any observable abnormal cervical pathology, the cervical low-grade squamous epithelial lesion group (LSIL group), the cervical high-grade squamous epithelial lesion group (HSIL group), and the cervical squamous cell carcinoma group (CA group). The remaining group served as the control, devoid of HPV infection.

All participants were married women within the reproductive age range. Notably, there were no statistically significant variations in mean age, number of births and other characteristics among the five patient groups (Table 1). This homogeneity effectively mitigates the influence of age and number of births on subsequent vaginal microbiota analyses.

Table 1 Characteristics of the study population.

Item		Group(%)	Total	H/χ2	p	
CA (n = 20)	HPV (n = 20)	HSIL (n = 20)	LSIL (n = 20)	N (n = 19)	
Age		44 (37, 52.25)	44.5 (40, 48.75)	45 (36.5, 54.75)	44.5 (35.25, 52.5)	45 (29, 52)	44 (36, 53)	0.789	0.94	
Production frequency		2 (1.250, 2.750)	2 (1, 2)	1.500 (1, 2.750)	2 (1, 2.75)	2 (0, 2)	2 (1, 2)	3.442	0.487	
Smoking status	Current smoker	13 (65)	10 (50)	14 (70)	9 (45)	11 (57.895)	57 (57.576)	3.481	0.481	
	Non-smoker	7 (35)	10 (50)	6 (30)	11 (55)	8 (42.105)	42 (42.424)			
Ctrual cycle	Luteal	8 (40)	6 (30)	7 (35)	7 (35)	7 (36.842)	35 (35.354)	1.171	0.997	
	Follicular	5 (25)	6 (30)	7 (35)	7 (35)	6 (31.579)	31 (31.313)			
	Unknown	7 (35)	8 (40)	6 (30)	6 (30)	6 (31.579)	33 (33.333)			
Contraception	Nil	7 (35)	6 (30)	5 (25)	6 (30)	3 (15.789)	27 (27.273)	3.825	0.873	
	Condoms	8 (40)	8 (40)	7 (35)	8 (40)	11 (57.895)	42 (42.424)			
	COCP	5 (25)	6 (30)	8 (40)	6 (30)	5 (26.316)	30 (30.303)			

Association between vaginal microbiota diversity and cervical carcinogenesis

To delve into the correlation between vaginal microbiota and cervical histological lesions in HPV-infected patients, we initiated 16S sequencing on vaginal samples from five patient cohorts. This allowed us to scrutinize the microbiota’s species structure and make a comparative assessment of the species richness of the vaginal microbiota.

Our α diversity analysis revealed a significant increase in the total number of vaginal microbiota species within the CA group compared to the other four groups (Fig. 1A). It is worth noting that in the other four groups, including the HPV group, precancerous lesion group, and non-infected control group (N group), there were no significant differences in total species numbers. Further enriching our understanding, the Shannon index analysis emphasized that the cervical cancer group exhibited increased vaginal microbiota species richness, including both total species count and evenness (Fig. 1B). Additionally, the species richness of the cervical high-grade squamous lesion group (H group) and the HPV group was slightly higher than that of the non-infected group and cervical low-grade squamous lesion group (L group), while there were no significant differences in species abundance between the H group and the HPV group, as well as between the non-infected group and the L group. These findings collectively illustrate that as cervical lesions induced by HPV infection intensify, the species richness of the vaginal microbiota amplifies—especially post the onset of HPV-induced cervical cancer.

Figure 1 Diversity analysis of vaginal microbiota in different groups.

(A) Alpha diversity analysis to compare the species diversity of vaginal microbiota among different groups of HPV-infected patients. (B) Shannon index analysis of species richness and species evenness of HPV-infected patients between different groups.

Alterations in vaginal microbiota composition accompanying cervical carcinogenesis

The noteworthy augmentation in species richness within the vaginal microbiota upon the onset of cervical cancer prompted us to explore whether concomitant changes occurred in its species composition. In the non-infected control group (N group), the predominant bacterial genus encompassed Lactobacillus, Gardnerella, Prevotella, and Streptococcus. In contrast, the vaginal microbiota composition in the four HPV-infected groups exhibited alterations primarily attributed to a reduction in Lactobacillus and an increase in Gardnerella (Fig. 2B), as compared to the N group. This finding signified a discernible decrease in Lactobacillus and an appreciable rise in Gardnerella proportions subsequent to persistent HPV infection.

Figure 2 Species composition of vaginal microbiota between different HPV-infected patient groups.

(A) Analysis of vaginal microbiota species composition, at the phylum taxonomic level. (B) Species composition analysis of vaginal microbiota, at the genus taxonomic level. (C) Dendrogram showing differences in species composition between groups, combined with vaginal microbiota species composition within groups.

This shift was more pronounced upon the development of cervical cancer, evidenced by a more substantial decline in Lactobacillus proportions compared to the other groups. Consistently aligned with the foregoing outcomes, at the phylum level, the abundance of bacteria in the Firmicutes phylum registered a substantial decrease within the cervical cancer group, while the prevalence of bacteria in the Actinobacteria phylum exhibited a concomitant increase (Figs. 2A and 2C). These cumulative findings suggest a potential association between Gardnerella bacteria and HPV infection, particularly in the context of cervical cancer.

Unveiling distinctive microbiota variations across groups

In our pursuit of discerning differences within the vaginal microbiota across the five patient groups, we engaged in a comprehensive beta diversity analysis. The outcomes bore an intriguing revelation: the vaginal microbiota within the CA group exhibited significant disparity from the other four groups, while the vaginal microbiota of these remaining four groups showed overall comparability (Fig. 3A). At the species level, we examined the similarity of vaginal microbiota subspecies among groups at the ASV level. Through PCoA analysis, we validated these results. The composition of vaginal microbiota in the CA group underwent significant changes. Additionally, compared to the non-infected group, the species composition of the other three groups also changed noticeably, although to a lesser extent than the CA group (Fig. 3B). These observations suggest that qualitative changes in the vaginal microbiota occur only in HPV-induced cervical carcinogenesis (Fig. 3B). These observations suggest that exclusive to HPV-induced cervical carcinogenesis, qualitative shifts manifest within the vaginal microbiota.

Figure 3 Comparison of differences in species diversity of vaginal microbiota between cervical cancer group and other groups.

(A) β diversity analysis reflects the degree of difference in vaginal microbiota between different groups. (B) PCoA analysis shows the similarity of vaginal microbiota subspecies between different groups at the level of ASV. (C) Lefse analysis revealed differences in vaginal microbiota composition between different groups. (D) Dendrogram showing differential microbiota among different groups.

Our exploration for specific bacterial disparities among the vaginal microbiota across different groups led us to employ Lefse analysis. This analysis highlighted the pronounced enrichment of Lactobacillus and Lactobacillus in the N group. Contrastingly, the CA group exhibited a notable rise in Veillonella. The Veillonella and Saccharopeptophilus bacteria were enriched in the non-infected control group (H group) and the HPV group, respectively (Figs. 3C and 3D).

Clinical validation indicating vaginal microbiota changes impact Th1/Th2 imbalance

Post HPV infection, the vaginal microbiota exhibited alterations, albeit not significantly profound. Remarkably, a substantial shift in the vaginal microbiota was evident solely after the development of HPV-induced cervical cancer (Fig. 4A), distinctively discernible from the non-cancerous group. We posited that these modified vaginal microbiota dynamics could be implicated in the initiation and progression of cervical cancer by regulating Th1/Th2 immune dysfunction. To scrutinize this hypothesis, we conducted a functional predictive analysis of the vaginal microbiota in each group. Intriguingly, the vaginal microbiota of the CA group displayed an association with ABC transporters.

Figure 4 Biological function analysis of vaginal micromicrobiota between different groups.

(A) Functional predictive analysis of genes expressed in vaginal microbiota between different groups. (B) Heat maps showing the correlation of specific bacterial groups with expression of inflammatory factors and age.

To gain further insights into the biological role of vaginal microbiota, we employed an Elisa test to assess the protein expression levels of Th1-related cytokines IL2, IL12, and Th2-related cytokines IL5, IL13, as well as the monocyte-macrophage-related factor TNFa in the centrifuged supernatant of vaginal-cervical secretions from diverse research subjects. Analyzing five inflammatory factors, our findings unveiled notable distinctions between the CA group and the other groups (Fig. 5). Correlation analysis unveiled that the enrichment of Lactobacillus exhibited a negative correlation with the expression of IL5/IL13 and TNFa, while positively correlating with the expression of IL2 and IL12. Conversely, Gardenerella demonstrated positive correlations with senescence and Th2 cytokine expression like IL5/IL13 and TNFa, and inversely correlated with Th1 cytokine expression such as IL2 and IL12 (Fig. 4B). These results suggest that these inflammatory factors might play a role in the onset and advancement of cervical cancer, influenced by the vaginal microbiota.

Figure 5 Variation analysis of five inflammatory factors between each groups.

(A) Five inflammatory factors between CA group and HPV group. (B) Five inflammatory factors between CA group and HSIL group. (C) A total of five inflammatory factors between CA group and N group.

Discussion

Cervical cancer is one of the most prevalent and fatal malignancies affecting women, primarily linked to HPV. HPV, a common clinical DNA virus, impacts human mucosal squamous epithelium and skin, serving as the primary risk factor for cervical cancer in women (Chelimo et al., 2013; Cohen et al., 2019). The female vagina constitutes a dynamic ecosystem, including microecological microbiota, vaginal anatomy, local endocrine functions, and immunity responses (Saraf et al., 2021). Under normal circumstances, the vaginal microecology of healthy women maintains a dynamic equilibrium with anti-infection capabilities, effectively hindering adherence and proliferation of exogenous substances and pathogens (The Integrative HMP (iHMP) Research Network Consortium, 2019). Any disruption to this balance increases the likelihood of exogenous pathogen infection, including HPV (Buchta, 2018). Persistent HPV infection has the potential to trigger cervical precancerous lesions and eventually cervical cancer. In our study, we explored the vaginal microbiota of HPV-infected patients with varying degrees of clinical cervical lesions. We observed that HPV infection induced changes in vaginal microbiota diversity, though not significantly. However, significant changes emerged in the vaginal microbiota of HPV-infected patients diagnosed with cervical cancer, particularly a reduction in Lactobacillus levels and an increase in Gardenerella presence.

The notable decline of Lactobacillus in the vaginal microbiota of cervical cancer patients is significant. Lactobacillus, once abundant in the vaginal microbiota of healthy women, is known for its immune-regulatory functions encompassing both innate and adaptive immunity. In innate immunity, vaginal epithelial cells express Toll-like receptors (TLRs) and Nod-like receptors (Mitra et al., 2016). These receptors trigger intracellular signaling pathways, leading to the release of cytokines like defensins, tumor necrosis factor (TNF)-α, interleukins (IL)-1, IL-6, IL-8, promoting an inflammatory response in the vaginal mucosa to combat pathogenic microorganisms (Wang et al., 2022). Additionally, Abramov et al. (2014) identified that, apart from Lactobacillus crispatus, Lactobacillus acidophilus and Lactobacillus jensenii can also induce the NF-κB pathway, regulating immune responses without inducing inflammatory reactions. Cellular immunity plays a crucial role in adaptive immune responses, with effector T cells activated to eliminate foreign viruses in the context of HPV infection (Moscicki et al., 2020), and the vaginal microbiota playing a role in regulating immune responses.

Existing literature extensively documents the dysregulation of TH-1 and TH-2 responses in the context of HPV persistent infection leading to cervical cancer (Clerici, Shearer & Clerici, 1998; Lin et al., 2019). Notably, the diminished expression of the Th1 cytokine IFNg has been linked to an unfavorable prognosis (Tartour et al., 1998). Additionally, deviations in the elevation of TH-1 cytokines have been tied to weakened treatment responses (Sharma et al., 2009). However, the link between this immune factor imbalance and vaginal microbiota has remained unexplored. We conducted an analysis to establish a connection between levels of TH-1 and TH-2 cytokines and specific prevalent bacterial groups. Our goal was to highlight the potential of modulating bacterial composition to impact TH-1 and TH-2 cytokine levels, reshaping the tumor microenvironment.

Our clinical validation and correlation analysis revealed a positive correlation between Lactobacillus and the expression of Th1 factors IL2/IL12. It is noteworthy that IL2/IL12 plays a pivotal role in the survival and activation of T lymphocytes, serving as a crucial component in anti-tumor responses (Lu, 2017). This suggests that in cervical cancer patients’ vaginal microbiota, there’s a reduction in bacteria beneficial for anti-tumor immunity, while the abundance of Gardenerella, which inversely correlates with the expression of inflammatory factors IL2/IL12, significantly increases. This increase in Gardenerella might impede tumor immunity, consequently facilitating cancer progression. Consistent with our research, prior studies have demonstrated that cervical cancer patients exhibit lower proportions of Th1 cells alongside higher proportions of Th2 and Treg cells compared to healthy controls. Elevated levels of IL-4, IL-10, and TGF-βI have been observed in the serum of cervical cancer patients, whereas INF-γ concentrations are diminished in this population (Lin et al., 2019). Our findings imply that the heightened expression of IL2/IL12 is linked to changes in the quantity and ratio of Gardenerella and Lactobacillus. This, in turn, suggests that targeted interventions to modulate the vaginal microbiota could potentially trigger the upregulation of Th1 factors’ expression, thereby fostering anti-tumor activity.

Moreover, in addition to the heightened expression of inflammatory factors IL2/IL12 in the CA group, our functional predictive analysis of the vaginal microbiota revealed an association with ABC transporters. Previous research has indicated that ABC transporters are linked to the adaptive immune system. The heterodimeric transporter (ABCB2/3) is associated with antigen processing TAP1/2 and constitutes a critical component of the adaptive immune system (Seyffer & Tampé, 2015). This ABC transport complex facilitates the transfer of proteasomal degradation products to the endoplasmic reticulum, where antigenic peptides are loaded onto major histocompatibility complex class I molecules and subsequently presented on the cell surface. Particularly noteworthy, during viral infections, the ABC trafficking complex participates in antigen presentation by cytotoxic T cells (Banerjee et al., 2023). As a result, it is plausible that in cases of HPV-induced cervical cancer, ABC transporters might play a role in tumor antigen presentation, thereby fostering anti-tumor immune responses.

We have introduced the pioneering notion that the vaginal microbiota could play a role in driving the advancement of cervical cancer by influencing the tumor’s immune microenvironment. Our investigation has also unveiled the predominant microbiota in the vaginal milieu of patients with varying lesions post HPV infection. However, it’s important to acknowledge that the limited sample size might warrant further validation of our conclusions. Additionally, the data presented here were gleaned from bioinformatic analyses of sequencing data. While corroborated by experimental evidence such as Elisa tests, the specific functions and mechanisms underpinning the alterations in vaginal microbiota among patients with HPV-induced cervical cancer require more in-depth exploration.

Conclusion

In summary, our study employed bacterial DNA sequencing and bioinformatics analysis on vaginal specimens from HPV-infected patients with varying degrees of lesions and control groups. While no significant variations were observed in the microbiota of distinct specimens, except for cervical cancer cases, we did ascertain a reduction in Lactobacillus levels and an increase in Gardenerella prevalence in HPV-induced cervical cancer. The functional predictive analysis of vaginal microbiota notably revealed the predominance of ABC transporters in cervical cancer patients. Elisa tests and correlation analyses further underscored the association between changes in vaginal microbiota and the production of Th1 factors IL2 and IL12. It is noteworthy that IL2 and IL12 are pivotal inflammatory cytokines intrinsic to tumor immunity. In light of these findings, our work underscores the potential involvement of vaginal microbiota in shaping the tumor immune microenvironment within cervical cancer, exerting an influence on tumor progression. Our research suggests that future treatments for cervical cancer could potentially involve manipulating the composition of the vaginal microbiota, such as increasing the abundance of Lactobacillus. This could be done to rectify an immune microenvironment skewed towards Th2 responses, shifting it towards a Th1 immune microenvironment. This modulation could, in turn, enhance anti- tumor immunity.

Supplemental Information

Supplemental Information 1 original data.

We would like to thank the researchers and study participants for their contributions.

Additional Information and Declarations

Competing Interests

Author Contributions

Human Ethics

Data Availability

The authors declare that they have no competing interests.

Yiheng Yang conceived and designed the experiments, prepared figures and/or tables, and approved the final draft.

Jufan Zhu conceived and designed the experiments, prepared figures and/or tables, and approved the final draft.

Renqian Feng performed the experiments, prepared figures and/or tables, and approved the final draft.

Mengfei Han performed the experiments, authored or reviewed drafts of the article, and approved the final draft.

Fenghua Chen analyzed the data, authored or reviewed drafts of the article, and approved the final draft.

Yan Hu analyzed the data, authored or reviewed drafts of the article, and approved the final draft.

The following information was supplied relating to ethical approvals (i.e., approving body and any reference numbers):

This study was conducted in accordance with the Declaration of Helsinki. The studies involving human participants were reviewed and approved by Professional ethics committee of clinical research of the First Affiliated Hospital of Wenzhou Medical University (Approval No. YS2018-058). The participants provided their written informed consent to participate in this study.

The following information was supplied regarding data availability:

The data are available in the Supplemental File.

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
