# Peer review of "Altered vaginal cervical microbiota diversity contributes to HPV-induced cervical cancer via inflammation regulation"

_PeerJ, doi:10.7717/peerj.17415_

## Round 0.1 · original submission · Minor Revisions

Dear Dr Yang,
Your manuscript has been reviewed by different reviewers. As you can see your manuscript requires revision as per the reviewers' suggestions.

Ravi Tandon

·

Basic reporting

No comment

Experimental design

No comment

Validity of the findings

No comment

Reviewer 2 ·

Basic reporting

1. Restriction of the background of the study in the abstract: Please restrict the background of the study incorporated in the abstract.
2. Sampling Location Omission: The manuscript lacks information on the specific locations from which the samples were collected. It is recommended to include these details in the abstract for comprehensive clarity.
3. Language Clarity and International Audience: The manuscript requires improvement in English language proficiency to enhance accessibility for an international audience. Instances of unclear language are noted in lines 53-54, 79-82, and Section 2.2. Consider involving a colleague proficient in English and knowledgeable in the subject, or consider professional editing services.
4. Biological Names Accuracy: Correctly write biological names to ensure accuracy; review and rewrite accordingly.
5. Figure Legends Revision: Rewrite figure legends accurately for clarity and precision.
6. Elaboration on Term "Defensins" (line 65): Provide a detailed explanation of the term "defensins" for better comprehension.
7. Remove the Last Line of the Introduction (89-91): Eliminate the last line of the Introduction (lines 89-91) for improved coherence.

Experimental design

1. Participant Information: Include the total number of participants approached, the number excluded, and the reasons for exclusion to enhance transparency.
2. Choosing women of reproductive age: What motivated the authors to select only women who were of reproductive age? Please justify.
3. Incorporate Significance Levels in Alpha and Beta Diversity: Verify and correct the biological names used in the manuscript for accuracy. Ensure adherence to proper nomenclature and rewrite as necessary.

Validity of the findings

1. Species-Level Alterations and Species Richness: While evaluating cervical and vaginal microbiome alterations in HPV-induced cervical cancer, the manuscript lacks mention of species-level changes. Additionally, though "species richness" is referred to, corresponding data is missing (line 167, section 3.2, Fig 1, Fig 2, Fig 3).
2. Rewriting the discussion After Species Data Incorporation: Following the inclusion of species data, the discussion section needs to be rewritten to accommodate the additional information.
3. Update References: Update references to ensure they are not older than 10 years, preferably within the last 5 years. This will align the manuscript with current literature and necessitate the rewriting of the introduction and discussion.

Reviewer 3 ·

Basic reporting

This article provides valuable insights into the complex interplay between vaginal microbiota, HPV infection, and cervical cancer development. This article investigates the association between vaginal microbiota, human papillomavirus (HPV) infection, and the development of cervical cancer, mainly focusing on the modulation of Th1/Th2 immune response. The article addresses a significant and timely issue, considering the rising prominence of cervical cancer and the known link with HPV infection. The study's methodology, involving the collection of vaginal samples from a substantial number of HPV-infected patients and control groups, as well as the use of bacterial DNA sequencing and cytokine quantification, demonstrates a comprehensive and rigorous approach. The article can be accepted by correcting minor grammatical errors and better-quality pictures.

Experimental design

Line 100: Exclusion and Inclusion criteria: The authors have decided upon certain criteria for their study it is unclear whether they have developed these criterions or modified an existing studies criterion to suit their study.

Figures:
Figure 2C, Dendrogram provided, is not clear; as it showcases the differences in species composition between groups, combined with vaginal microbiota better quality picture should be used

Validity of the findings

Line 140: 2.3.2 (2)16S rRNA-V4 region amplification: References for PCR reaction parameters are missing if it’s unclear whether the authors have developed it or not.

Additional comments

The wording in the discussion could be slightly improved to eliminate the unnecessary use of dangling words.

---

## Round 0.2 · Minor Revisions

Dear author

As you can see, one of the reviewers has raised concerns. Please address his comments.

Regards

Ravi Tandon

Reviewer 2 ·

Basic reporting

No comments

Experimental design

Incorporate individual Significance Levels in Alpha and Beta Diversity graphs and also in the text.

Validity of the findings

While evaluating cervical and vaginal microbiome alterations in HPV-induced cervical cancer, the manuscript lacks mention of species-level changes. The experimental data about the change in the abundance of different species of Lactobacillus should be clearly mentioned in the results section.

Additional comments

No comments

Reviewer 3 ·

Basic reporting

The article is having sufficient introduction and background to demonstrate how the authors have designed study / hypothesised. Review of literature is appropriately referenced

Experimental design

The design of study seems fitting, the sampling, inclusion and exclusion criteria are clear. Methodologies adopted from previous literature are properly referenced. The authors have meticulous efforts in designing the stud

Validity of the findings

The statistical findings are sound, robust and Controlled. The conclusions are appropriately stated.

---

## Round 0.3 · accepted · Accept

Dear Dr. Yang,

Thank you for your submission to PeerJ.

I am writing to inform you that your manuscript - Altered Vaginal Cervical Microbiota Diversity contributes to HPV-induced cervical cancer via inflammation regulation - has been Accepted for publication. Congratulations!